# Superfluorescence of Sub-Band States in C-Plane In_0.1_Ga_0.9_N/GaN Multiple-QWs

**DOI:** 10.3390/nano12030327

**Published:** 2022-01-20

**Authors:** Cairong Ding, Zesheng Lv, Xueran Zeng, Baijun Zhang

**Affiliations:** State Key Laboratory of Optoelectronic Materials and Technologies, School of Electronics and Information Technology (School of Microelectronics), Sun Yat-sen University, Guangzhou 510275, China; lvzsh@mail2.sysu.edu.cn (Z.L.); zengxr@mail.sysu.edu.cn (X.Z.); zhbaij@mail.sysu.edu.cn (B.Z.)

**Keywords:** InGaN/GaN multiple-quantum wells, superfluorescence, time-resolved photoluminescence, photoluminescence, collective emissions

## Abstract

Superfluorescence is a collective emission from quantum coherent emitters due to quantum fluctuations. This is characterized by the existence of the delay time ($\tau_{D})$ for the emitters coupling and phase-synchronizing to each other spontaneously. Here we report the observation of superfluorescence in c-plane In_0.1_Ga_0.9_N/GaN multiple-quantum wells by time-integrated and time-resolved photoluminescence spectroscopy under higher excitation fluences of the 267 nm laser and at room temperature, showing a characteristic $\tau_{D}$ from 79 ps to 62 ps and the ultrafast radiative decay (7.5 ps) after a burst of photons. Time-resolved traces present a small quantum oscillation from coupled In_0.1_Ga_0.9_N/GaN multiple-quantum wells. The superfluorescence is attributed to the radiative recombination of coherent emitters distributing on strongly localized subband states, *E_e_*_1_→*E_hh_*_1_ or *E_e_*_1_→*E_lh_*_1_ in 3nm width multiple-quantum wells. Our work paves the way for deepening the understanding of the emission mechanism in the In_0.1_Ga_0.9_N/GaN quantum well at a higher injected carrier density.

## 1. Introduction

Spontaneous emissions (SEs) of photons occur because of coupling between excited two-level systems and vacuum modes of the electromagnetic field, such as the process of fluorescence that is commonly used in displays and lighting—effectively stimulated by its zero-point fluctuations. C-plane InGaN/GaN quantum wells (QWs) as the prevailing active layer have been studied extensively due to their promising applications in group-III nitride semiconductor optoelectronic devices [1,2,3,4]. Commonly accepted explanations for emission features are the spatial localization of carriers due to random alloy fluctuations, indium compositional fluctuations and well width fluctuations [5,6,7,8,9,10,11,12,13,14,15,16,17,18], the quantum-confined Stark effect (QCSE) because it spatially separates electron and hole wave functions and reduces the wave function overlap in the QWs [19,20,21,22,23,24,25] and the screening of the QCSE under a high excitation that affects the excitation density-dependent emission energy of InGaN/GaN MQWs, for example, a very strong emission from quantum-dot-like states [26], high energy emission band [27,28,29] and stimulated emission on the high-energy side in thick QWs [30,31,32].

Rime-resolved photoluminescence (TRPL) presents fast and slow decay behaviors over several orders of magnitude due to weakly-localized carriers transferring among different energy levels easily [33,34]. The decay dynamics of carriers at a time scale longer than 200 ns is ascribed to agglomerated localization centers on nearly isolated islands [35].

Light–matter interactions can also create and manipulate collective many-body phases in solids. Lasing and cooperative emissions (superradiance SR and superfluorescence SF) form a family of collectively coherent emissions—a collective emission that results in a burst of photons with ultrafast radiative decay. SR was described by Dicke in 1954 [36] followed by the experimental observation in 1973 [37]. Several SF emissions were reported [38,39]. SR and SF are characterized by an accelerated radiative decay time τ ∝ $\tau_{SE}$/N, where the exponential decay time $\tau_{SE}$ of SE from the uncoupled two level systems is shortened by the number of coupled emitters N [40,41]. In addition, the peak intensity of SF and SR increases superlinearly with N, $I_{SF}\propto N^{2}$.

SF emissions are typically characterized by the unique existence of the delay time $\tau_{D}\propto \log\left(N\right)/N$ for the development of macroscopic spontaneous quantum coherence due to quantum fluctuations [36,37,38] and Burnham–Chiao ringing which can’t be observed in all experiments primarily due to a spatial averaging effect [42] and pure superfluorescence [41].

In this case, we observe a previously unreported unique behavior of the SF associated with the uniformity of localized states in In_0.1_Ga_0.9_N/GaN QWs especially at high excitation level with which generally, the manufactured devices like laser diodes are always operating [43]. Hence, it is necessary to deepen the understanding of the emission mechanism at high excitation levels.

## 2. Materials and Methods

InGaN/GaN MQWs schematized in Figure 1a were grown on (0001) c-plane sapphire substrates in a metal organic chemical vapor deposition (MOCVD) system with the Thomas Swan closely spaced showerhead reactor. The In content in the 3 nm thick InGaN QW is 10%. The measured period of the multiple QW system is about 13 nm, leading to 10 nm barrier width.

Time-integrated and time-resolved photoluminescence measurements were performed at room temperature using a regeneratively amplified femtosecond Ti: sapphire laser with a repetition rate of 1 kHz operating around 800 nm. The 25 fs pulses duration were tripled to provide a 267 nm excitation source with the maximum power of about 20 mW (Micro-5 + Legend Elite USP HE+, Coherent), corresponding to $2.686\times 10^{9}$ photons per pulse.

The excitation laser output was reflected to a prism in order to obtain a 267 nm laser. The 267 nm laser beam was in alignment with the center height of the slit by way of two mirrors and apertures and focused by a singlet lens with the focal length of 70 mm onto the sample In_0.1_Ga_0.9_N/GaN MQWs in the way of an oblique incidence. The diameter d of the laser spot is 1.488 × 10^−4^ cm, the area of the laser spot is 1.738 × 10^−8^ cm^2^. We were thus able to reach the maximum excitation densities of 1150.8 kW cm^−2^. The backward emission from the sample was collected perpendicular to the sample surface by a duplet lens system (f = 70 mm and 320 mm, d = 70 mm) and converged onto the input slit of a double-grating monochromator (SpectraPro hrs-300, PI) and detected by a CCD camera (Optronis SRU-ED, Kehl, Germany) which are connected to an ultraviolet-visible streak camera system with 2 ps time resolution and an instrument response function of 6 ps (Optoscope SC-10 SRU-ED, Optronis GmBH, Kehl, Germany). The 300 mm spectrograph has two gratings with 300 and 1200 lines per mm. The input slit of the spectrograph is a vertical slit; while the output slit of the streak camera is a horizontal one. We adjusted slit width to 10 μm. The time base is 0~288 ps under the fastest sweep speed of 15 ps/mm. The pump power (P) is controlled with a variable neutral density filter.

## 3. Results and Discussion

Figure 1 presents the excitation density-dependent photoluminescence (PL) spectra of In_0.1_Ga_0.9_N/GaN MQWs for the excitation fluence (P) from 256.9 to 985.1 kW cm^−2^. The emission spectra were measured in the range of 2.6–3.1 eV. At the low excitation less than 347.6 kWcm^−2^ (Figure 1b), the emission exhibits a flat and broad PL spectrum line shape with full width at half maximum (FWHM) of about 277 meV (inset, Figure 1c top panel), which is the convolution of two emission peaks at the center energy 2.763 eV and 2.922 eV. These two emissions are attributed to sequential spontaneous emissions of shallow impurities [44]. However, at the excitation density of 444.8 kW cm^−2^, a narrow emission shoulder peak of 2.945 eV appears at the high energy side in PL. As excitation fluence is increased, the emission peak energy is blue-shifted up to 2.960 eV at the exaction density of 661.7 kW cm^−2^.

The quantitative analysis of spectrally integrated PL which is best fitted with a Lorentzian for various excitation power densities is shown in Figure 1c (the inset shows the fitting to the spectrum at the excitation fluence 815.3 kWcm^−2^, bottom panel). As the excitation fluence is increased from the excitation density of 461.6 kW cm^−2^, PL peak energies first present the blueshift feature (top panel). That will be discussed in detail in Figure 2. As the excitation fluence is further increased from 704.8 kW cm^−2^ to 985.1 kW.cm^−2^, the emission peak energy is red-shifted from 2.960 eV to 2.949 eV presumably owing to renormalization of the emission energy from the dense coherent coupling. The peak intensity increases superlinearly over three orders of magnitude (middle panel) according to a power-law dependence with an exponent of α = 2.4 ± 0.1 deviating from the theoretically expected value of α = 2 [42] presumably by assuming that more localized states devote to nonlinear emissions. At the excitation density exceeds 985.1 kW.cm^−2^, the emission peak intensity reduces due to the horizontal transmission of the delocalized carriers [45] and renormalization of the emission energy. Delocalized carriers face a higher number of fast non-radiative recombination centers leading to an increase of non-radiative losses. The full-width at half-maximum (FWHM) of the PL spectrum is excitation sensitive and reduced to about 30 meV after the excitation threshold (bottom panel). The emission-line narrowing is one of the observed features of the SF [39]. Remarkably, narrow emission lines are similar to the 36 meV linewidth measured on a single GaN quantum dot at room temperature [46]. These narrow contributions are, therefore, fully compatible with the expected emission from a single localized state.

Figure 2 discussed the blueshift of the emission peak energy. Generally, the PL peak energy of the InGaN/GaN MQWs can be expressed as [47]:(1)$$E_{p}=E_{g}+\Delta E_{strain}-\Delta E_{QCSE}+\Delta E_{QSE}$$
where $E_{g}$ is the bandgap of relaxed InGaN alloys, $\Delta E_{strain}$ represents the blueshift caused by the compressive strain, $\Delta E_{QCSE}$ represents the well width dependent red shift in the effective band gap due to the quantum confined Stack effect (QCSE) [48], and $\Delta E_{QSE}$ represents the shift caused by the intersubband transition due to the quantum size effect (QSE). In our InGaN MQWs, the first two items would not be changed by the excitation power. Thus, the $\Delta E_{QCSE}$ and $\Delta E_{QSE}$ could be responsible for the blue-shift of the PL spectrum as the excitation fluence increases. As we know, the QCSE usually causes red-shifts owing to the tilt of energy bands. Moreover, under a high excitation fluence, the QCSE may be weakened by the huge number of photogenerated carriers, resulting in a decrease of QCSE-caused red-shift, i.e., a PL peak blue-shift, However, this process does not result in a time-delayed blueshift emission peak, as observed in Figure 3d. Thus, the discussed blue-shift is more likely caused by $\Delta E_{QSE}$, the changes in the intersubband transitions.

To clarify the mechanism in the observed cooperative emission and related peak shifts, the energy band diagram of the structure was simulated at 0 V, using Silvaco TCAD software, as shown in Figure 2a. Besides, a band detail of the QW is illustrated as Figure 3b in which the subband energies with different *n* were calculated using:(2)$$E_{n}=\frac{\hbar^{2}\pi^{2}}{2m^{*}L_{w}^{2}}n^{2}$$
where ℏ is the reduced Planck constant, *L_W_* is the well width, and effective masses of In_0.1_Ga_0.9_N were taken in the calculations are: *m_e_* = 0.21 *m*_0_ for electrons, *m_lh_* = 1.22 m_0_ for light holes and *m_hh_* = 2.00 *m*_0_ for heavy holes [49]. The results indicate that the subband energies in the conduction band are more discrete owing to the smaller effective mass of electrons. The energy difference between *E_e_*_1_ and *E_e_*_2_ is about 0.597 eV, far exceeding the measured PL peak variation of 15 meV. Thus, we deduce that the transitions from *E_e_*_1_ to *E_lhn_* or *E_hhn_* should be dominant in our experiments. In the case of 267 nm excitation (*hυ _=_* 4.662 eV), photogenerated holes distribute the multiple subbands of the valence band, and then two processes will happen synchronously: (1) photogenerated holes at high energy levels will relax to the lowest *E_hh_*_1_ gradually; (2) radiative recombination occurs between photogenerated electrons at *E_e_*_1_ and photogenerated holes at multiple subband energies. As the excitation power increases, more and more holes will relax to the lowest *E_hh_*_1_ before they are recombined with electrons. Therefore, at the excitation fluence 461.6 kW cm^−2^, the carriers’ transition from *E_e_*_1_ to *E_hh_*_1_ enhances gradually, leading to the increasing PL intensity at 2.945 eV and a narrow FWHM. Meanwhile, the delay time is observed as the excitation density increases due to the coherent emitters (Figure 3b–f), also supporting the relaxation process of the holes. At slightly higher excitation fluence, the states at *E_hh_*_1_ will be completely filled by photogenerated and relaxed holes owing to their limited state density. Thus, the transition from *E_e_*_1_ to *E_hh_*_1_ tends to saturation, and the transition from *E_e_*_1_ to *E_lh_*_1_ dominates gradually, leading to the emission peak blueshift for the excitation fluence from 461.6 to 704.8 kW cm^−2^. Furthermore, the delay time will decrease gradually with the excitation fluence (Figure 3g, lower panel) because coherent emitters from *E_e_*_1_-*E_lh_*_1_ transitions form earlier spontaneously than coherent emitters from *E_e_*_1_-*E_hh_*_1_ transitions. In addition, the calculated *E_lh_*_1_ is about 0.013 eV higher than *E_hh_*_1_, which also agrees with the observed blue shift from 2.945 eV to 2.960 eV. Above all, the observed narrow emissions should come from the localized intersubband transitions of the InGaN wells, in which SF from *E_e_*_1_-*E_hh_*_1_ transition is dominant at the excitation fluence of 461.6 kW.cm^−2^ and SF from *E_e_*_1_-*E_lh_*_1_ coherent transitions will be enhanced with an increasing excitation fluence. Thus, the blueshift of the emission from the QWs is confirmed to originate from the confined subband levels when the well width is smaller than 3–4 nm [44].

Figure 3 presents streak camera images and extracted time-resolved emission intensity traces (over 100 meV energy window) for various excitation densities in detail and quantitative analysis. At the low excitation fluence of 344.1 kW cm^−2^, two emission peak energies at 2.820 and 2.941 eV exhibit almost the same dynamics for spontaneous emissions of shallow impurities (Figure 3a). The rise time of the emission peak energy at 2.941 eV (blue trace, Figure 3d) is 8ps shorter than that of the emission peak energy at 2.820 eV (red trace, Figure 3d), showing carriers competitively populate on shallow impurity energy levels. The initial time of these two emissions is almost the same as that of the emission of GaN top surface. With P = 461.6 kW cm^−2^, the peak emission energy shows the blueshift to 2.945 eV and its dynamic process presents the distinct delay time $\tau_{D}$ before the rising edge and a more shortened radiative decay and long radiative decay (green trace in Figure 3b,d). As the excitation fluence of 661.7 kW cm^−2^, in addition to the $\tau_{D}$, we cannot simply describe the time-resolved PL as the bi-exponential decay (Figure 3c and purple trace, Figure 3d).

Figure 3e,f show extracted time-resolved emission traces for different excitation fluences over 100 meV energy window. In time-resolved PL decay measurements with low excitation fluence less than 444.8 kW cm^−2^ (yellow trace in Figure 3e), we did not observe a significant modification in the decay of the spontaneous emission, which can be described as a momo-exponential decay with 1/e decay time of $\tau_{SE}$ = 1180 ps.

At a slightly higher excitation fluence 461.6 kW cm^−2^ (cyan trace in Figure 3e), after the delay time $\tau_{D},$ we observed a more pulse named SF with 1/e decay time of $\tau_{SF}$ = 67 ps in comparison to the PL decay of SE with 1/e decay times of $\tau_{SE}$ = 1180 ps. The delay time ($\tau_{D})$ was calculated as the period between the start point of the emission and the rising edge of 2.945 eV. As the excitation densities *P* increase up to 524.3 kW cm^−2^ (Figure 3f), after the $\tau_{D}$, a rising edge and the accelerated PL decay of the emission with a moderately long tail cannot be described by a mono-or bi-exponential function of the spontaneous emission, as shown in Figure 3c. As excitation fluences increase more and more, such as at 815.3 kW cm^−2^ (Figure 3h, lower panel), the time-resolved emission intensity trace displays a small oscillatory superfluorescence shape [41,42] during 118~165 ps, which is different from traces fitted by mono-exponential (red line) and bi-exponential(blue line) decay functions. However, a pure superfluorescence pulse has a single hyperbolic-secant shape [41]. Since the penetrating depth of laser has effects on the distribution of state densities of a high-gain medium, When the laser incidents on c-plane In_0.1_Ga_0.9_N/GaN multi-QWs, the sample may thus be considered as being divided into a number of concentric shells of decreasing density, the coupling of carriers in multi-QWs’ layers by diffraction makes spatial averaging which washes out the ringing pulse period.

As excitation fluence increases further, the decay time ($\tau_{SF})$ shortens to 8 ps (upper panel in Figure 3g), exceeding the instrument response limit by 6 ps at FWHM. The cyan line presents that $\tau_{SF}$ is fitted to the excitation fluence *P* for excitation densities over 461.6 kW cm^−2^, $\tau_{SF}\propto$1180 α/*P*, where α is a scaling factor. The initial number of coupled emitters *N* is proportional to the excitation fluence P/α. Here, a fixed value of $\tau_{SE}$ = 1180 ps was used, as found for spontaneous emission of In_0.1_Ga_0.9_N/GaN QWs at low excitation density. The proportional relation is in accord with $\tau_{SF}\;\propto \;\tau_{SE}/N$ of cooperative emissions [36,41]. The lower amplitudes of slower components are observed (Figure 3c). Consequently, the SF decay rate should converge towards the decay rate of spontaneous emission.

Furthermore, a shortening of the SF delay time (lower panel in Figure 3g), after which the photon burst is emitted, is observed. This characteristic of SF is a consequence of the time it takes for the individual dipoles to become phase-locked and scales with the number N of excited coupled emitters according to $\tau_{D}\;\propto$ log(*N*)/*N* [38,39,41]. The data are fitted to $\tau_{D}$ = 46 + 5313 ×
*log*(*P*)/*P* (blue line, Figure 3g lower panel) by assuming N to be proportional to the excitation fluence *P*. The drastic shortening of the radiative lifetime and the delay time attest to the observed emission being superfluorescence. The delay time $\tau_{D}$ decreases for high excitation fluences owing to the increased interaction among emitters, which is in accordance with the red-shift of the emission peak energy for higher excitation fluences discussed in Figure 1c, upper panel.

Figure 3h,i, present the decay image of the emission peak energy of 2.967 eV with the excitation density of 815.3 kW cm^−2^ and dynamic red-shift slices integrated over a 5 ps time window at different times delays in the semi-logarithmic scale. Although the observed image of SF is alike to the Fermi-edge SF in In_0.2_Ga_0.8_As/GaAs [50], as time goes on, the dynamic red-shift in our In_0.1_Ga_0.9_N/GaN is attributed to the coherently radiative emission of carriers distributed on E*_e_*_1_ and E*_hh_*_1_, carriers reduce during the relaxation (Figure 2b). It takes much more time for photogenerated holes at high energy levels to relax to the lowest *E_hh_*_1_ gradually and form coherent emitters, resulting in a longer $\tau_{D}$. SF coherent states result from these narrow localization states. In addition, a dynamical red-shift is owing to the renormalization of the emission energy from the coherent coupling. The SF mission maybe undergoes the evolution process of the coherent emission to the noncoherent emission due to the large lateral dimensions or number fluctuations within the coherent SF state.

Figure 4 presents streak camera images of SF emission for four typical excitation fluences and spectrally integrated time-resolved PL traces (over 100 meV energy window) for six typical excitation fluences from 317.9 to 871.2 kW cm^−2^ in a new batch of the sample. In Figure 4a, the build-up evolution of the SF photon burst is clearly shown on the base of the delay time $\tau_{D}$. At the *P* = 473.3 kW cm^−2^, SF appears clearly after the delay time. At the *P* = 871.2 kWcm^−2^, a brighter image after the $\tau_{D}$ is shown due to photon bursts of SF. Figure 4b presents normalized time-resolved PL traces for different fluences in order to observe the evolution of the $\tau_{D}$ and the decay trace of the emission more clearly. As the excitation fluence increase, $\tau_{D}=26+4490\times \log\left(P\right)/P\;$(Figure 4b, inset), which confirms to $\tau_{D}\;\propto$ log(N)/N of SF [38,39,41]. After photon bursts of SF, the decay trace of the emission shortens much with the increase of excitation fluences. In Figure 4c, after the overlap (the black arrow) of decay traces, the emission intensity increase a little bit (red traces, the magnified image in green circle of the inset) at the excitation fluence of 698.5 kW cm^−2^, which means some weak oscillation is observed in the time domain compared with the decay trace under the weak excitation fluence of 317.9 kW cm^−2^ (black trace). The 3 nm QWs’ width may contribute to the SF, Quantum confined localization states are coupled with QWs, which results in the oscillation. The oscillation is a fundamental property of SF [42]. Figure 4d shows the fitting to fluorescence traces of SE and SF at excitation fluences of 317.9 kW cm^−2^ and 769.9 kW cm^−2^ respectively. The small oscillation was considered to be ignored due to error problems during the fitting. Decay traces are simply described by a bi-exponential function, there are no damped oscillations involved. The rising edge of the SE is described as a biphasic dose response function given that one part of carriers distribute on upper levels and another part of photons may start to form cooperative mode because the end of the rising edge shows the characteristics of the formation process. The rising edge of SF was described as a revised Double Boltzmann function, or a polynomial function given that the SE and the formation of cooperative states occur in this period. The horizontal transmission was observed with the side geometry (not shown there). These observations were consistently reproducible. In addition, the output face of the sample was not limited to a small spatially-resolve region, thus, the remarkable oscillation was not demonstrated in our experiments at room temperature. The strong SF ring is observed in 1–10 μm lateral dimensions of a homogeneous system [38], or a small detect region [41,51] at low temperature. The output of the sample as a whole is the sum of the radiation from each of the small dimensions, and this “spatial averaging” washes out the ringing or oscillation.

Given the previous observations, our experiments revealed that the SF from the highly excited In_0.1_Ga_0.9_N/GaN QWs at room temperature can be confirmed unambiguously by the evidence for the characteristic delay time $\tau_{D}$, $\tau_{SF}\propto \tau_{SE}/N$, emission line narrowing with the increasing of excitation fluences, and quadratic *P* dependence of the peak emission intensity $I_{SF}$. This is attributed to the coherent radiative emissions of carriers distributing on the subband *E_e_*_1_ and *E_hh_*_1_ or *E_lh_*_1_ of localized states in In_0.1_Ga_0.9_N/GaN MQWs at higher injected carrier density. The oscillation of the emission pulse was not observed clearly owing to spatial averaging and room temperature. The results opened a door for us to research collectively emissive characteristics of quantum wells. The fitting to the SF pulse further demonstrated its characteristics.

As the SF critically depends on the low decoherence and low inhomogeneous energy variance, the long dephasing time ($T_{2}^{*}$) [52] which is mainly dependent on the stringent quality of the material structure, high oscillator strength, and small inhomogeneous broadening. In the time domain, $T_{2}^{*}$ can be directly measured with free-induction decay, optical nutation, photon-echo, etc. SF occurs when $\sqrt{\tau_{SF}\tau_{D}}<T_{2}^{*}$. The higher the pump pulse energy is, the shorter $\tau_{D}$ is. Thus, the dephasing is limited by the pump pulse energy. In our experiment, the value of $\sqrt{\tau_{SF}\tau_{D}}$ = 22 ps ($\tau_{SF}$ = 8 ps and $\tau_{D}$= 62 ps in Figure 2g), which is in the range of $T_{2}^{*}$ about 10–50 ps typical for high quality bulk semiconductors [53].

As the excitation fluence further increases, the growth rate of the field becomes high enough to exceed the dephasing rate and SF emission develops. The linewidth initially drops with increasing pump intensity and becomes saturated once SF becomes dominant.

However, in condensed matter systems, SF has been difficult to observe due to the inherently short coherence times of carriers. The SF in In_0.2_Ga_0.8_As/GaAs QWs was attributed to many-body renormalization and Coulomb enhancement of gain [54]. Strongly localized states of the subband of In_0.1_Ga_0.9_N/GaN QWs have been demonstrated in a few nm carrier localization domains and electrons travel at most over a scale of a few tens of nm [55]. It can be reasonably inferred that the single localized state spontaneously develops into macroscopically coherent emitters due to the high oscillator strength, which devotes to observing SF at room temperature. In the following work, we will study the coherent process from the polarization characteristics of the emission. Under strong excitation, the interaction among multi-bodies in quantum wells deserves research.

## 4. Conclusions

Our measurements reveal that coherent SF coupling presents in In_0.1_Ga_0.9_N/GaN quantum well due to strongly localized states at high excitation fluence and room temperature. This opened up new opportunities to deepen the understanding of the emission mechanism in In_0.1_Ga_0.9_N/GaN quantum wells as the prevailing active layer of optoelectronic devices with high-brightness at a higher injected carrier density, and could enable the exploitation of cooperative effects for optoelectronic devices and then prompted us to study the coherent polarization characteristics and the coherent dephasing process of emitters by four-wave-mixing [56], transient resonant four-wave mixing [57], and the second-order coherence of the emission [38]. The interaction among multi-bodies in In_0.1_Ga_0.9_N/GaN quantum wells and other semiconductor quantum wells should still be a research branch due to their wide applications for a large variety of optoelectronic devices.

## Figures and Tables

**Figure 1 nanomaterials-12-00327-f001:**
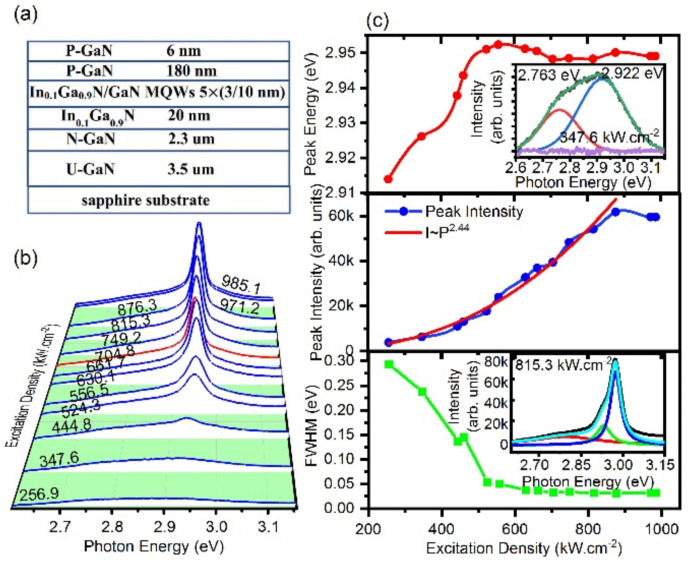
(Color Online) Excitation fluence dependent photoluminescence behaviors of ${\mathrm{In}}_{0.1}{\mathrm{Ga}}_{0.9}N/\mathrm{GaN}$ MQWs (**a**) Schematic diagram of material structure. (**b**) Color Mapping of PL intensities and the emission photon energy at different excitation fluences (P) from 256.9 kW.cm^−2^ to 985.1 kW.cm^−2^. (**c**) Top, the shift of the peak emission energy with the excitation fluence. The inset exhibits the flat and broad PL spectrum line at low excitation fluence; Middle, the peak emission intensity that increases superlinearly with the excitation fluence, corresponding to a power-law dependence with an exponent α = 2.4 (red line); Bottom, the narrowing of the FWHM with the excitation fluence obtained from the peak fitting to the spectra in (**b**). The inset shows several typical spectra. Different color lines in subfigures (**c**) present fitting peaks to spectra with a Lorentzian for various excitation power densities.

**Figure 2 nanomaterials-12-00327-f002:**
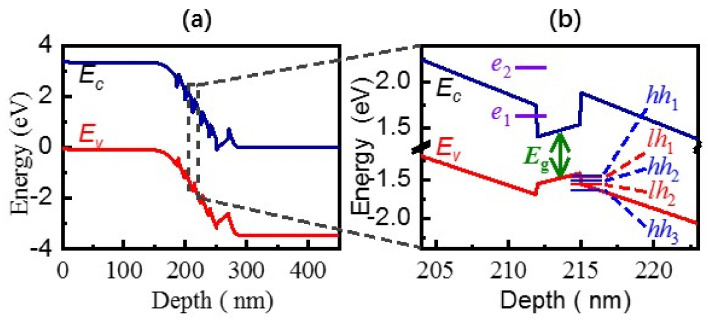
(Color Online) (**a**) Energy band diagram simulated by Silvaco TCAD software at 0 V bias; (**b**) energy band details of the InGaN QWs, together with the calculated subband energies.

**Figure 3 nanomaterials-12-00327-f003:**
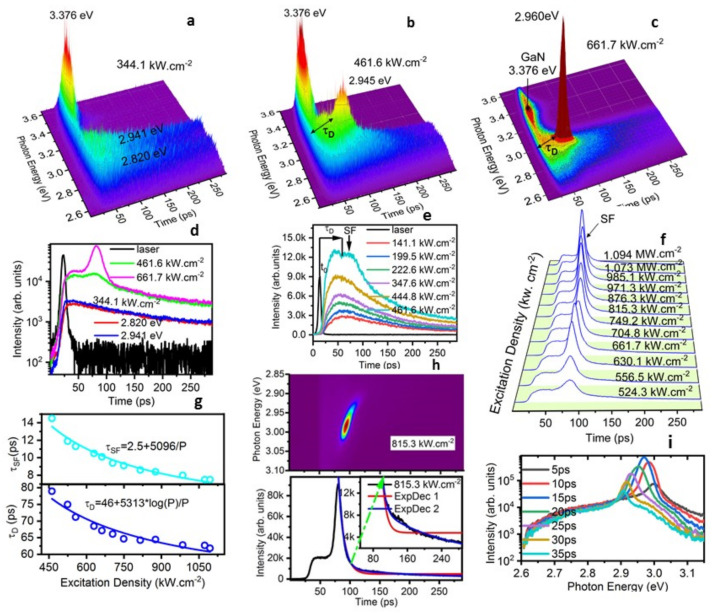
(Color Online) Steak camera images and time-resolved photoluminescence spectroscopy for different excitation fluences. (**a**) weak excitation 344.1 kW.cm^−2^. (**b**) Observation of delayed bursts of radiation with threshold excitation 461.6 kW cm^−2^. (**c**) Strong excitation 661.7 kW cm^−2^. The delay time ($\tau_{D}$) is defined (**d**) Time-resolved photoluminescence spectroscopy for three excitation fluences above. The laser given here shows the initial time of the emission. Obviously, the delay emission is observed after the threshold excitation. (**e**,**f**) Extracted time-resolved emission intensity traces for different excitation fluences over 100 meV energy window in detail. The initial time is indicated as t_0_. $\tau_{D}$, the delay time; SF, superfluorescence. (**g**) Upper, decay time ($\tau_{SF}$) as a function of the excitation fluence (*P*), fitted using the superfluorescence model (cyan line): $\tau_{SF}=2.5+5096/P$. (**h**) Upper, streak camera image obtained at an excitation fluence of 815.5 kW cm^−2^. Lower, time-resolved emission intensity trace (with an enlarged view) presents the definite difference from traces fitted by ExpDec 1 (red line) and ExcDey 2 (blue line) functions. (**i**) The dynamic redshift of time-resolved slices integrated over a 5 ps time window at different time delays on a semi-log scale.

**Figure 4 nanomaterials-12-00327-f004:**
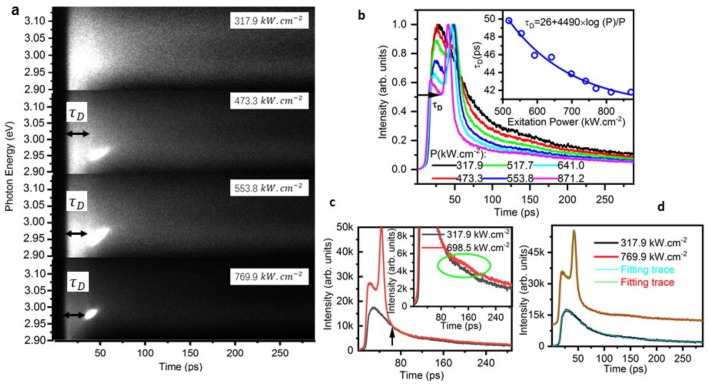
(Color Online) Superfluorescence from In_0.1_Ga_0.9_N/GaN QW. (**a**) Streak camera images of SF emission for four typical excitation fluences, 317.9 kWcm^−2^, 473.3 kWcm^−2^, 553.8 kWcm^−2^, 769.9 kW.cm^−2^, the $\tau_{D}$ means the delay time. (**b**) Spectrally integrated time-resolved PL traces (over 100 meV energy window) for different excitation fluences (P units: kW.cm^−2^). Inset, the excitation fluence of the delay time was best fitted with a function $\tau_{D}=26+4490\times \log\left(P\right)/P$. (**c**) Time-resolved PL traces for excitation fluences of 698.5 kW cm^−2^ (red) compared to the time-resolved PL trace at 317.9 kW cm^−2^ (black), the insets show a small ringing circled in the green ellipse, the black arrow points at the overlap of the falling edge of emission traces. (**d**) The fitting to fluorescence traces of SE and SF at excitation fluences of 317.9 kW.cm^−2^ and 769.9 kW.cm^−2^ respectively.

## Data Availability

Data underlying the results presented in this paper are not publicly available at this time but may be obtained from the authors upon reasonable request.

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
