# Peer review of "Superfluorescence of Sub-Band States in C-Plane In0.1Ga0.9N/GaN Multiple-QWs"

_nanomaterials, 2022, doi:10.3390/nano12030327_

Round 1

Reviewer 1 Report

The paper is interesting and describes the photophysical characterization of QW materials and observed superfluorescence emission due to their collective optical response.

The study is well described, but the clarity of the presentation/figures should be improved before formal acceptance.

A figure like 4b has too many colors and numbers. It's not easy to follow. I suggest splitting data into two figures or making one larger figure to make the story more readable (easier to follow).

Fig 2 d should be corrected - text within the figure must be visible/readable.

Additionally, the quality of the fit of fluorescence decays is missing in the text. It must be added so that referees and readers can judge the quality of the measurements. 

The conclusion must be enlarged, and the authors should highlight/discuss significant findings in more detail.

In the end, I am recommending a minor but obligatory revision of the draft.

Author Response

Cover letter

Greetings referee,

I am very glad to receive these valuable comments and suggestions.  

I have revised those relevant contents of my manuscript on base of your suggestions. The responses are as following:

Response 1:  I have revised the whole paper carefully

Response 2: I corrected some fine/minor spelling in English language and style, which is shown as red words

Response 3: In the revised version, the results have been presented clearly

 “Given the previous observations, our experiments revealed that the SF from the highly excited In0.1Ga0.9N/GaN QWs at room temperature can be confirmed unambiguously by the evidence for the characteristic delay time ,  , emission line narrowing with the increasing of excitation fluences, and quadratic P dependence of the peak emission intensity . This is attributed to the coherent radiative emissions of carriers distributing on the subband Ee1 and Ehh1 or Elh1 of localized states in In0.1Ga0.9N/GaN MQWs at higher injected carrier density. The oscillation of the emission pulse was not observed clearly owing to spatial averaging and room temperature. The results open a door for us to research collectively emissive characteristics of quantum wells. The fitting to the SF pulse further demonstrated its characteristics.”

Response 4: The conclusions can be supported by the results in the revised version

Response 5: The presentation/figures has been improved

            I simplified the Fig. 4b by giving 6 typical emission traces from initial 14 emission traces

            The text within the Fig. 2d was adjusted to be visible/readable

            The fit of fluorescence decays has been added in the revised version. I hope to give some reasonable explanations.

Response 6: The conclusion has been enlarged

“Our measurements reveal that coherent SF coupling presents in In0.1Ga0.9N/GaN quantum well due to strong localized states at high excitation fluence and room temperature. This opened up new opportunities to deepen the understanding of the emission mechanism in In0.1Ga0.9N/GaN quantum wells as the prevailing active layer of optoelectronic devices with high-brightness at a higher injected carrier density, and could enable the exploitation of cooperative effects for optoelectronic devices. and then promoted us to study on the coherent polarization characteristics and the coherent dephasing process of emitters by four-wave-mixing, transient resonant four-wave mixing, and the second-order coherence of the emission. The interaction among multi-bodies in In0.1Ga0.9N/GaN quantum wells and other semiconductor quantum wells should still be a research branch due to their wide applications for a large variety of optoelectronic devices.”

Best Regards

Ding Cai Rong

Reviewer 2 Report

The manuscript is devoted to the measurements of the time-integrated and time-resolved photoluminescence. The measurements concerned In0.1Ga0.9N/GaN multiple-quantum wells systems grown on the sapphire substrate. The primary Authors’ result was observing the superfluorescence effects at room temperature, which were thoroughly described in the article. The observed effects are related to the presence of strongly localized states in a high excitation regime. The Authors emphasized that the presented results can help deepen the understanding of the emission mechanism in discussed quantum well systems. The results seem to be correct and valid enough to be published. 

The manuscript is well written in general and is accessible to a broad range of Readers. The Authors provide a comprehensive list of references that could be a good starting point for the studies in the field. All results are presented clearly and properly discussed with numerous references to other positions in the literature.

Author Response

Cover letter

Greeting Referee,

I am very glad to receive these valuable comments and suggestions.  

I have revised some minor spellings and improved some contents in the revised version. I hope to present the physical essence. I will continue to study on the coherent characteristics in semiconductor quantum wells due to their prevailing applications for optoelectronics devices.

I hope you can sign your review report.

Best Regards

Ding Cai Rong
